# Microstructure and Corrosion Properties of Laser Cladding Fe-Based Alloy Coating on 27SiMn Steel Surface

**Changyao Ouyang [1], Qiaofeng Bai [1], Xianguo Yan [1,*], Zhi Chen [1], Binhui Han [2] and Yan Liu [3]**

1   School of Mechanical Engineering, Taiyuan University of Science and Technology, Taiyuan 030024, China; ouyangcy123@163.com (C.O.); baiqiaofeng1@163.com (Q.B.); mechenzhi@tyust.edu.cn (Z.C.)
2   School of Aviation Maintenance Engineering, Xi'an Aviation Vocational and Technical College, Xi'an 71008, China; hbhzayy@126.com
3   Shanxi Design and Research Institute of Mechanical and Electrical Engineering Co., Ltd., Taiyuan 030009, China; liuyang123vi@126.com
*   Correspondence: yanxianguo@tyust.edu.cn

**Abstract:** In this paper, the corrosion performance of a laser cladding Fe-based alloy coating on the surface of 27SiMn steel was studied. The Fe-based alloy coating was prepared on a 27SiMn steel surface by high-speed laser cladding. The microstructure, morphological characteristics, element content, and phase composition of the cladding layer were analyzed by an optical microscope (OM), scanning electron microscope (SEM), energy dispersive spectrometer (EDS), and X-ray diffractometer (XRD), respectively. The corrosion resistance of the 27SiMn substrate and Fe-based coating in different corrosive environments was tested through an electrochemical experimental station, a salt spray corrosion test box, and an immersion experiment. The Fe-based alloy cladding layer is mainly composed of a-Fe, $M_7C_3$, $M_2B$, and $Cr_3Si$. The cladding layer structure forms planar, cellular, dendrite, and equiaxed dendrite during rapid solidification. The corrosion potential of the cladding layer is higher than that of the substrate, and the arc radius of the cladding layer is larger than that of the substrate. After salt spray corrosion, a large number of red and black corrosion products appeared on the surface of the substrate; the surface of the cladding layer sample was still smooth, and the morphology was almost unchanged. The weight loss results of the cladding layer and 27SiMn matrix after 120 h of immersion are 0.0688 and 0.0993 g·cm$^{-2}$, respectively. The weight loss of the cladding layer is 30.7% less than that of the matrix. Conclusion: Laser cladding an Fe-based alloy coating on the surface of 27SiMn has better corrosion resistance than the substrate, which improves the corrosion resistance of hydraulic supports.

**Keywords:** laser cladding; Fe-based alloy coating; 27SiMn steel; microstructure; electrochemical corrosion; salt spray corrosion; immersion corrosion

## 1. Introduction

The 27SiMn hydraulic support is indispensable core support equipment for underground mining. To adapt to the complex environment of coal mines, the support column not only needs to have high enough hardness, wear resistance, impact resistance, and other properties but also needs to have good corrosion resistance. To effectively improve the corrosion resistance of the support column, it can be surface modified to improve its corrosion resistance. Laser cladding technology is a kind of material surface modification technology. Its principle is to melt the metal powder and weld it on the substrate surface with a high-energy laser beam so that the substrate and cladding surface can achieve good metallurgical bonding [1].

In recent years, a large number of scholars have studied laser cladding. Saeedi et al. [2] prepared NiCr–TiC metal matrix composites by laser cladding NiCr–TiC and NiCr–TiC powders on a stainless-steel matrix to improve the hardness and corrosion re-

sistance of AISI420 (Martensitic stainless steel) stainless steel. Farotade et al. [3] studied the microstructural characteristics and surface properties of a Ti–6Al–4V alloy laser cladding nickel–zirconium–boron coating.

For the main material of hydraulic support, the laser cladding of 27SiMn steel, related researchers have also conducted a lot of research. Chai et al. [4] studied the effect of laser power on the laser cladding of iron-based alloys on the surface of 27SiMn steel, and the results showed that the tensile strength and microhardness of the cladding layer prepared under different laser power levels were higher than that of the 27SiMn steel substrate. Yang et al. [5] used JG-2(A specially developed iron-based alloy powder brand) and JG-3(A specially developed iron-based alloy powder brand), two kinds of iron-based alloy cladding powders, for cladding on 27SiMn hydraulic support. By comparing the contents of the two kinds of powder elements, they found that JG-2 alloy powder contained more Cr and Ni elements, was easier to form a passivation film, and could strongly hinder the development of corrosion.

The hydraulic prop is used downhole, and water droplets from the formation are sprayed on it to form a film of water on the surface. The water film on the surface of the hydraulic prop contains strong corrosive anions, causing electrochemical corrosion reactions [6]. The corrosion of metals in the atmosphere is mainly affected by the atmosphere, the corrosion composition, and the corrosion factors of pollutants [7]. The middle and upper parts of the hydraulic support work in the complex environment of the underground coal mines, and the middle and upper parts are mainly corroded by the atmosphere. The lower part of the hydraulic prop is immersed in mine water with weak acid and a large number of anions in the water quality.

In this paper, an Fe-based alloy coating was prepared by high-speed laser cladding on the surface of 27SiMn steel, and the microstructure, phase composition, and element composition of the coating were analyzed. The corrosion resistance of the substrate and coating under different corrosion environments was analyzed by electrochemical corrosion, salt spray corrosion, and immersion corrosion.

## 2. Materials and Methods

### 2.1. Material

The substrate used in this test is a hollow cylindrical 27SiMn material with an outer diameter of 123 mm and a length of 600 mm. Sanding with sandpaper was performed before cladding to remove rust and oil until the surface showed a metallic luster. It was then cleaned with alcohol and dried for use. Its chemical composition is listed in Table 1.

**Table 1.** Chemical composition of the laser cladding matrix 27SiMn (wt%).

| C | Si | Mn | V | Ni | Cu | S | P | Fe |
|---|---|---|---|---|---|---|---|---|
| 0.24–0.32 | 1.10–1.40 | 1.10–1.40 | 0.07–0.12 | ≤ 0.30 | ≤ 0.30 | ≤ 0.04 | ≤ 0.04 | Bal |

The cladding powder used is an Fe-based alloy powder with an average diameter of 38 μm. The spherical shape of the powder can ensure better flow properties. It is composed of Fe, Cr, Ni, Si powder, and other trace element powders. The specific ingredients are shown in Table 2.

**Table 2.** Chemical composition of laser cladding powder (wt%).

| C | Cr | Si | Ni | B | Mo | Fe |
|---|---|---|---|---|---|---|
| 0.1 | 19.7 | 1.06 | 2.69 | 1.00 | 0.47 | Bal |

## 2.2. Preparation of the Cladding Layer

Use the RFL (Raycus Fiber Laser)-C4000 high-power fiber laser with the DPSF (Dual Package System Framework)-2 powder feeder was used to perform synchronous powder feeding laser cladding on the substrate, and the shielding gas and powder feeding gas were both Ar. The specific cladding process parameters are shown in Table 3.

**Table 3.** Laser cladding parameters.

| Parameters | Value |
|---|---|
| Power/W | 2700 |
| Powder feeding rate/(rad·min$^{-1}$) | 2.8 |
| Cladding rate/(m·min$^{-1}$) | 9 |
| Defocus/mm | 15 |
| Overlap rate (%) | 80 |

## 2.3. Method

The sample after laser cladding was cut into multiple pieces of $10 \times 10 \times 10$ mm$^3$ using a wire-cutting machine. The surface and cross-section of the cladding layer were ground and polished. The metallographic structure of the sample section was observed by a Motic-AE2000MET optical microscope (Motic, Hong Kong, China) after metallographic corrosion. A Zeiss (ZEISS: ΣIGMA300) field emission scanning electron microscope (Carl Zeiss, Jena, Germany) was used to observe the microstructure and morphology of the sample, and the element distribution and content of the coating were detected by an energy spectrometer (EDS, Carl Zeiss, Jena, Germany). An Empyrean X-ray diffractometer (PANalytical, Almelo, The Netherlands) was used to determine the phase of the sample. The test conditions were a Cu target, the diffraction range was 20–80°, and the diffraction speed was 2°/min. The diffraction data were analyzed and processed using Jade6.5. A VH-Z100R microscope (KEYENCE, Osaka, Japan) was used to observe the surface corrosion morphology.

Electrochemical experiments were carried out on an electrochemical workstation with an RST5000 three-electrode system (Ceres, Zhengzhou, China). The samples were treated in advance. In the electrochemical corrosion test, 3.5% NaCl solution was used as the electrolyte solution, the sample with the test area of 1 cm$^2$ was used as the working electrode, the saturated calomel electrode was used as the reference electrode, and the metal platinum sheet was used as the auxiliary electrode. Under the condition of stable open-circuit potential, the test frequency range of the test was set to $10^{-2}$–$10^5$ Hz and the AC (Alternating current) amplitude to 10 mV for the impedance test. A scanning rate of 0.5 mV/s was set to conduct the action potential polarization test in the range of −1000–2000 mV.

The LJ-60A salt spray corrosion box continuous spray was used for the salt spray experiment test (simulating atmospheric corrosion). The sample was treated before the salt spray test. The spray pressure was maintained at 1 kg/cm$^2$, the temperature in the salt spray box was 35 ± 2 °C, and the corrosive medium was 5 ± 0.5% NaCl aqueous solution. The salt spray experiment lasted for 48 h.

The static corrosion of the cladding layer and substrate was evaluated by an immersion test. The corrosive liquid simulated mine water was mixed with 5% (NaCl, CaSO$_4$, FeSO$_4$) in a ratio of 1:1:1, and sulfuric acid was added to mix the corrosive liquid with a pH value of about 3. Before soaking, the sample was pretreated and weighed, and placed into the sample bottle. They were then placed into a corrosive solution for 120 h and removed. After the taken-out sample was cleaned and air-dried, the mass of the sample was weighed with an electronic balance with an accuracy of 0.0001 g.

## 3. Results and Discussion

The cross-sectional morphology of the cladding layer is shown in Figure 1. It can be seen that the cladding layer is intact, without pores and cracks.

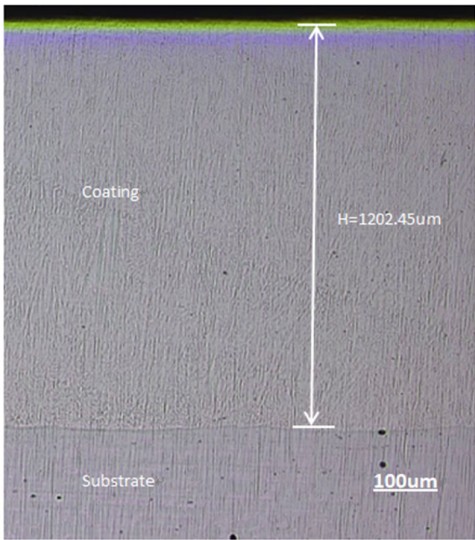

**Figure 1.** Thickness of the cladding layer and cross-sectional micromorphology

### 3.1. Phase Analysis

Figure 2 shows the XRD pattern of the laser cladding Fe-based alloy cladding layer. It can be seen from Figure 2 that the Fe-based alloy cladding layer is mainly composed of a-Fe solid solution and intermetallic compounds $M_7C_3$, $M_2B$, $Cr_3Si$. The lattice parameters of a-Fe in the body-centered cubic structure (bcc) of the cladding layer are $a$ = 0.2886 nm, as measured by the experiment. Compared with the lattice parameter of body-centered cubic a-Fe at room temperature ($a$ = 0.2866 nm) [8], the lattice parameter of a-Fe of the cladding layer increased. With the change in the lattice parameters, the same group of parallel (hkl) crystal planes will be affected, and the interplanar spacing $d_{hkl}$ will also increase. The calculation formula is as follows [9]:

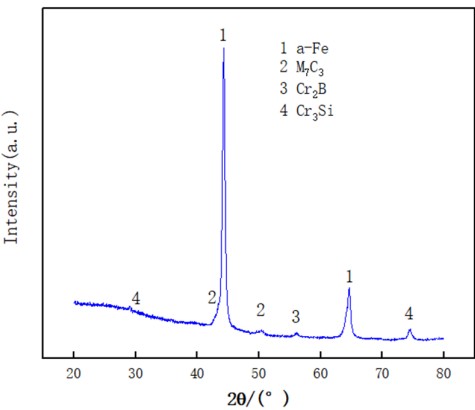

**Figure 2.** XRD pattern of the Fe-based alloy cladding layer.

The relationship between the distance between crystal planes and its index in the cubic crystal system is:

$$d = \frac{a}{\sqrt{h^2 + k^2 + l^2}} \tag{1}$$

where $a$ is the lattice parameter; $h$, $k$, and $l$ are the crystal plane indices; and $d$ is the crystal plane spacing.

This is mainly due to the different properties of solute atoms and solvent atoms and different atomic radii. When the solute atoms dissolve into the crystal lattice of the solvent, the lattice parameters of the solid solution will increase. C, B, and Cr in the alloy powder melt into the solvent metal (Fe) crystal lattice to form a solid solution. There is no obvious $\gamma$-Fe phase residual in the cladding layer, which is mainly caused by the very high cooling rate during the laser cladding process. Under such a high cooling rate, the austenite enters the martensite transformation region without growing up [10–12].

$M_7C_3$ and $M_2B$ are metal interstitial compounds formed by the transition metal M with a larger radius and C and B with a smaller radius in the alloy powder. $Cr_3Si$ is a metal close-packed phase formed by Cr atoms and Si atoms.

*3.2. Microstructure Morphology of the Coating*

The structure of the cladding layer formed planar crystals, cell crystals, dendrites, and equiaxed dendrites during rapid solidification. It can be seen from Figure 3a that the structure at the bottom of the cladding layer grows in a planar crystal manner and forms a good metallurgical bond with the matrix. As the cladding thickness increases and the cladding process changes, the temperature of the base material is low. The undercooling degree of the structure at the bottom of the cladding layer is small, and the temperature gradient G is large. Due to the existence of a temperature gradient and concentration gradient, although the supersaturation is uniform, the crystal will still grow into a flat crystal plane. This situation is because the growth rate of the entire crystal plane is determined by the supersaturation of the solute concentration at the outcropping point on the crystal plane, which is the core point that plays a dominant role and promotes growth on the crystal plane [13]. At a large temperature gradient G and a small crystal growth rate R, the G/R value is large, so the solidified structure grows in a flat plane at a low speed to form a flat crystal. In the area above the plane crystal, a new interface morphology composed of many convex circular cells similar to the paraboloid of revolution- and network-shaped recessed grooves appears. This morphology is called cellular crystal [14]. This is because the distance between the region and the substrate increases, and the temperature gradient–solidification rate ratio decreases, resulting in the formation of a cellular crystal structure. Bartkowski et al. [15] also came to a similar conclusion. He reported that the temperature gradient–solidification rate ratio decreased, resulting in the formation of the cellular crystal structure.

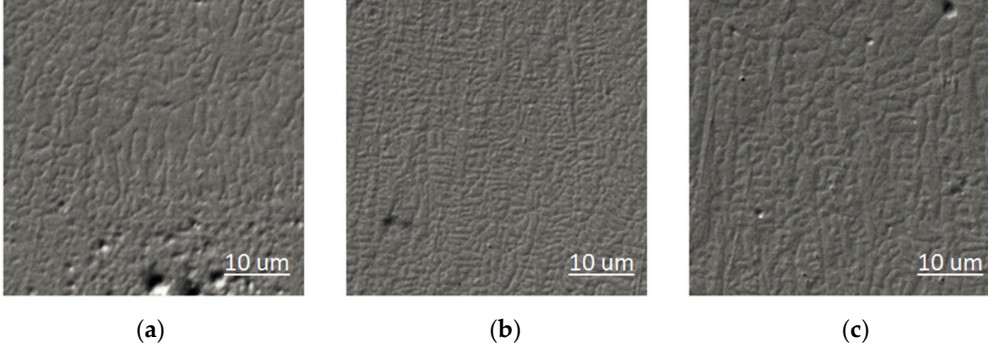

|  (a)  |  (b)  |  (c)  |

**Figure 3.** Cross-section structure of Fe-based alloy laser cladding layer on the 27SiMn surface: (**a**) bottom of the cladding layer; (**b**) middle of the cladding layer; (**c**) upper part of the cladding layer.

It can be seen from Figure 3b that the middle of the cladding layer is a columnar dendrite structure. The growth direction of cellular crystals is perpendicular to the solid–liquid interface and has nothing to do with crystallographic orientation. The G/R (Temperature gradient G, solidification rate R) ratio decreases, and the solute concentration changes. At this time, the convex unit cell extends farther to the melt, facing the new component undercooling; the original parabolic interface of the cell crystal gradually becomes unstable. The growth direction of the unit cell begins to turn to the preferential crystal growth direction. The lateral direction of the cell crystal will also be affected by crystallographic factors, and a flange structure will appear. When the composition is supercooled and strengthened, a sawtooth structure will appear on the flange (that is, columnar dendrites). It can be seen from Figure 3c that the dendritic structure of the cladding layer is refined in the upper surface area of the cladding layer [16]. The grain size is smaller than the size of the cladding layer at the bottom and the middle. This is because the upper part of the cladding layer is easier to contact with the outside air; the heat loss is accelerated, and the component supercooling zone is widened [17]. The small temperature gradient G and high cooling rate v make new crystal nuclei appear in the liquid at the same time or continuously, prevent the unidirectional extension of dendrites, and promote the formation of small equiaxed branches on the upper part of the cladding layer.

Figure 4 shows an EDS diagram of the microstructure in the middle region of the prepared cladding layer. It can be seen that the content of Fe, Cr, and Ni metal elements is consistent with the powder content, indicating that although segregation occurs during the crystallization process of the cladding layer structure, no significant segregation and deposition of metal elements occurs in the cladding layer area. Figure 5 shows an EDS analysis diagram at the interface of the cladding layer. It can be seen that the Fe element content increases from the cladding layer to the substrate, while the Cr and Ni metal element content decreases. This is mainly because the content of the 27SiMn substrate iron element is higher than that of the cladding layer. The Cr and Ni content of the layer is higher than that of the substrate. The results of EDS composition analysis show that the melting of part of the matrix in the transition area will cause the elements in the matrix to enter the molten pool, resulting in the dilution of the elements in the cladding layer, and the closer to the matrix, the more obvious the dilution phenomenon.

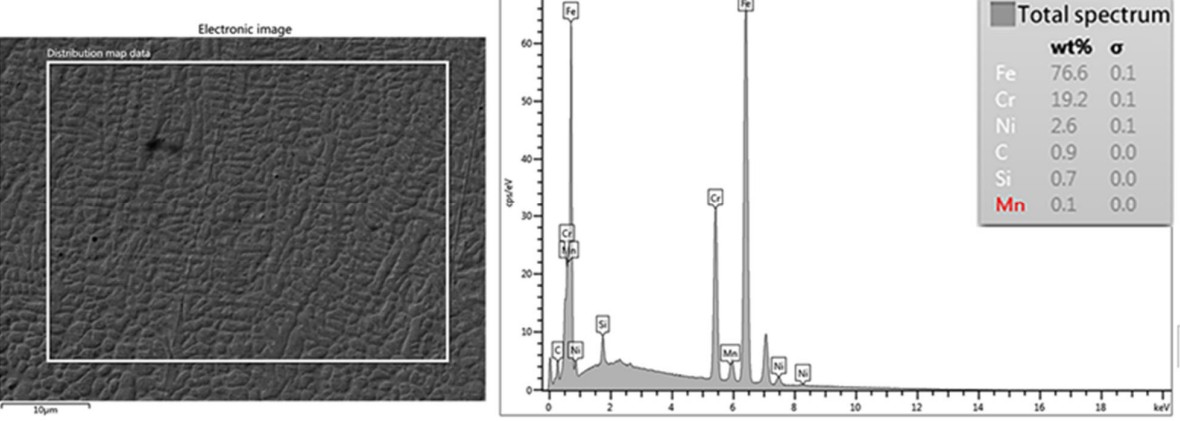

**Figure 4.** EDS analysis diagram of the microstructure in the middle area of the cross-section of the Figure 5. EDS analysis diagram at the interface of Fe-based alloy laser cladding layer.

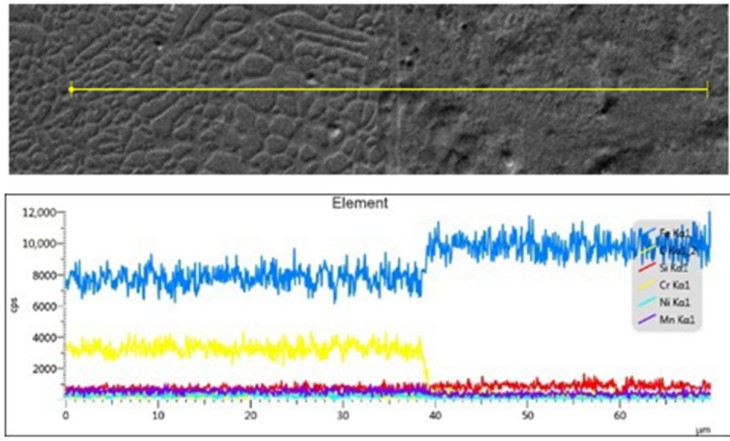

**Figure 5.** EDS analysis diagram at the interface of Fe-based alloy laser cladding layer.

### 3.3. Electrochemical Corrosion Performance

The electrochemical impedance spectroscopy (EIS) of the cladding layer and the substrate in 3.5% NaCl solution is shown in Figure 6. It can be seen from Figure 6a that the capacitive arc radius of the cladding layer is larger than that of the substrate, which indicates that the cladding layer has better corrosion resistance. Laser cladding an Fe coating on the surface of 27SiMn can improve the corrosion of hydraulic props.

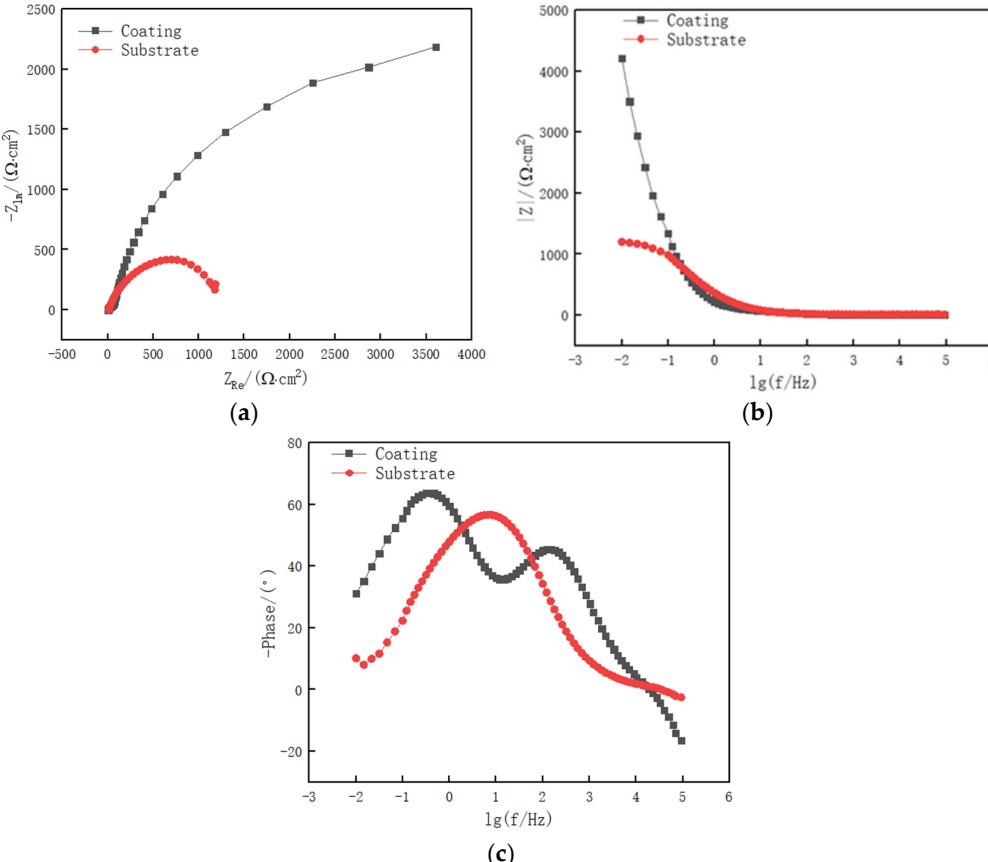

**Figure 6.** EIS diagram of the 27SiMn matrix and Fe-based alloy cladding layer in 3.5% NaCl solution: (**a**) Nyquist; (**b**) Bode of impedance; (**c**) Bode of the phase angle.

It can be seen from the Bode plot in Figure 6b that the impedance mode |Z| value of the cladding layer in the low-frequency band is greater than that of the 27SiMn matrix. According to Formula (2):

$$|Z| = \sqrt{Z'^2 + Z''^2} \tag{2}$$

where $Z'$ is the real part of the impedance, and $Z''$ is the imaginary part of the impedance. The impedance model of the cladding layer is 4.22 × 103 ohm, and the impedance model of 27SiMn is 1.20 × 103 ohm. The impedance modulus value in the low-frequency region represents the degree of penetration of the solution into the sample. The low-frequency impedance modulus value of the cladding layer is significantly greater than that of the substrate, indicating that the solution penetration of the cladding layer is less than the substrate, and the cladding layer is resistant to electrochemical corrosion Ability is better than the matrix.

It can be seen from Figure 6c that with the change of frequency, the phase peak of the 27SiMn matrix and the cladding layer is also constantly changing. In the low-frequency and high-frequency phases, the phase peak of the cladding layer is larger than the substrate; at the intermediate frequency phase, the phase peak of the cladding layer is smaller than the substrate. The small phase peak has high capacitance performance, and more charge accumulation is easily generated on the surface, which accelerates the corrosion of the sample [18,19]. Compared with the substrate, the cladding layer corrodes slowly at low and high frequencies and corrodes faster in the middle-frequency region.

Figure 7 shows the Tafel polarization curve of the 27SiMn substrate and Fe-based alloy cladding layer in 3.5% NaCl solution. The corrosion potential of the cladding layer is −0.2503 V, and the corrosion potential of the 27SiMn substrate is −0.3479 V. The corrosion potential reflects the difficulty of the material being corroded. The smaller the corrosion potential, the easier it is to be corroded [20]. The cladding layer has a higher corrosion electric potential and is more resistant to corrosion than 27SiMn.

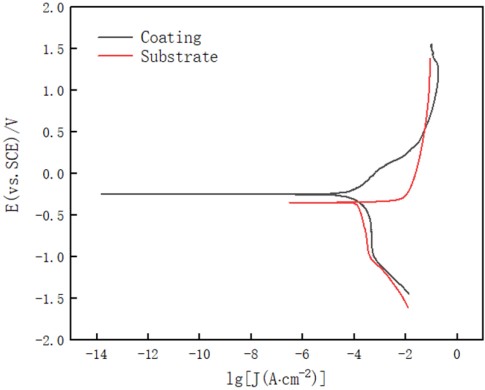

**Figure 7.** Tafel polarization curve of the 27SiMn substrate and Fe-based alloy cladding layer in 3.5% NaCl solution.

Figure 8 shows the electrochemical corrosion morphology of the cladding layer and 27SiMn substrate. It can be observed from the figure that the surface of the cladding layer undergoes pitting corrosion and local corrosion after electrochemical corrosion, and the surface of the substrate undergoes surface corrosion after electrochemical corrosion. The cladding layer and 27SiMn matrix are in NaCl solution, the cathode reaction is a reduction reaction, and the reaction process is as follows:

$$O_2 + 2H_2O + 4e = 4OH^- \tag{3}$$

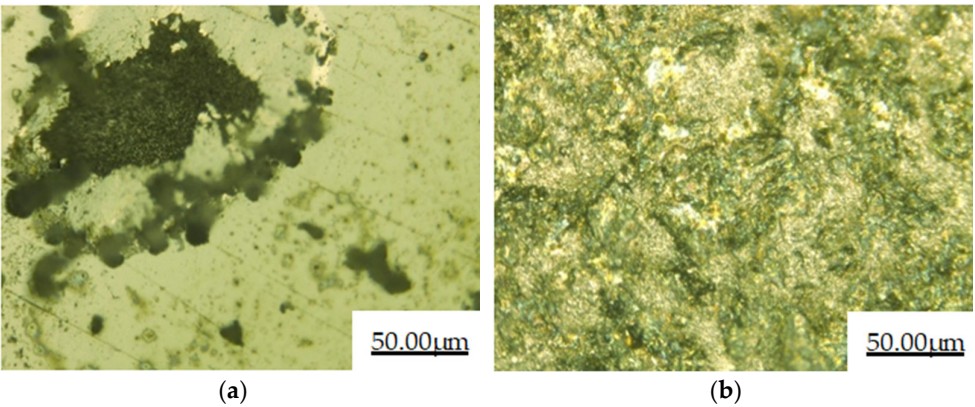

|  |  |
|:---:|:---:|
| (**a**) | (**b**) |

**Figure 8.** Electrochemical corrosion morphology of the (**a**) cladding layer and (**b**) 27SiMn substrate.

The cladding layer and 27SiMn substrate anode are oxidation reactions, and the reaction process is as follows:

$$\text{Cladding layer: } Fe - 2e = Fe^{2+} \tag{4}$$

$$Ni - 2e = Ni^{2+} \tag{5}$$

$$Mo - 3e = Mo^{3+} \tag{6}$$

$$\text{27SiMn matrix: } Fe - 2e = Fe^{2+} \tag{7}$$

$$Mn - 2e = Mn^{2+} \tag{8}$$

According to Table 4 [21], it can be seen that the cladding layer is more resistant to corrosion than the substrate. This is mainly because the alloy powder has Ni and Mo elements, which have a higher reduction potential than Fe. Although the equilibrium potential of Cr is much lower than that of iron, Cr is easily passivated. It improves the corrosion potential of the cladding layer. The reduction potential of Mn in the 27SiMn matrix is lower than that of Fe.

**Table 4.** Standard reduction potential of main metals $\Phi 0$ (v, vs, SHE-Standard hydrogen electrode), 25 °C.

| Electrode Reaction | $Mn = Mn^{2+} + 2e$ | $Cr = Cr^{3+} + 3e$ | $Fe = Fe^{2+} + 2e$ | $Ni = Ni^{2+} + 2e$ | $Mo = Mo^{3+} + 3e$ |
|---|---|---|---|---|---|
| Standard reduction potential $\Phi 0$ | −1.18 | −0.74 | −0.440 | −0.250 | −0.2 (about) |

*3.4. Salt Spray Corrosion Performance*

Figure 9 shows the salt spray corrosion morphology of the cladding layer and 27SiMn substrate. It can be observed that a large number of red and black rusts appear on the surface of the substrate, and the corrosion is more serious. However, the surface of the cladding sample is still flat and smooth, and the morphology is almost unchanged. Macroscopically, it shows that the cladding layer has excellent salt spray corrosion resistance. The relationship between the number of alloying elements and the corrosion rate (v) of steel is as follows [22,23]:

$$V = 6.44 - 1.167\,(C) - 1.001\,(Si) - 0.269\,(Mn) - 2.410\,(S) - 10.812\,(P) - 0.744\,(Ni) - 2.438\,(Cu) - 1.182\,(Cr) \tag{9}$$

Using the relationship between the alloying element content and the corrosion rate in Formula (9), according to the average chemical composition content of Fe alloy powder and 27SiMn, the calculated corrosion rate of the cladding layer is smaller than that of the 27SiMn substrate. This is mainly because the Cr and Si elements of the cladding layer significantly improve the atmospheric corrosion resistance of the cladding layer, while

the increase in Mn content has no obvious effect on the atmospheric corrosion resistance of the steel [22–24]. The results are similar to the atmospheric corrosion results of 27SiMn reported by Junyan et al. [23]. Cr is very easy to passivate and is very stable under natural conditions and many corrosive media. Cr is not only easy to passivate under conditions containing oxidants and oxygen but can also passivate in water. The dense passivation film of the cladding layer can effectively prevent oxygen in the air from penetrating the cladding layer. At the same time, the passivation film can be repaired by itself after being damaged in the atmospheric environment. Si helps prevent intergranular corrosion while cladding the intermetallic compounds carbides, borides, and Cr3Si formed in the coating layer. It can effectively reduce the penetration channel of the corrosive medium Cl⁻ and prevent the corrosive medium from penetrating the cladding layer. Improve the atmospheric corrosion resistance of the cladding layer. Lin et al. [25] also pointed out that chromium passivation protects the coating and improves its corrosion resistance.

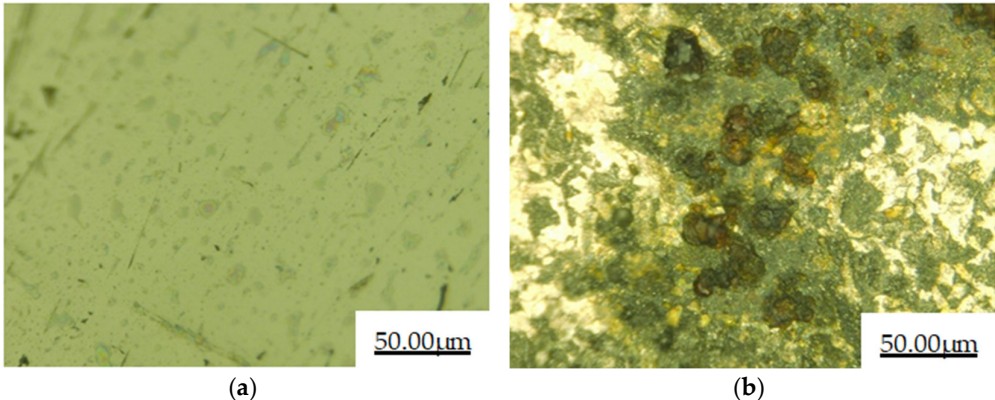

(**a**)        (**b**)

**Figure 9.** Salt spray corrosion morphology of the (**a**) cladding layer and (**b**) 27SiMn substrate.

*3.5. Immersion Corrosion Performance*

Figure 10 shows the immersion corrosion morphology of the cladding layer and 27SiMn substrate in the corrosive solution simulated mine water for 120 h. According to the corrosion morphology, it can be seen that pitting corrosion has occurred in the cladding layer, and uneven ulcer corrosion occurred in the substrate. Groups of pits, which are difficult to count, occurred on the surface of the cladding layer. This is mainly due to the destruction of the passivation film and oxide film formed on the cladding layer under long-term immersion corrosion [26]. The pit-like local corrosion of different sizes and depths appeared on the surface of the substrate. The weight loss of the cladding layer and 27SiMn substrate after 120 h of immersion was, respectively, 0.0688 and 0.0993 $g \cdot cm^{-2}$. It can be seen that the weight loss of the cladding layer during immersion corrosion is less than that of the substrate. This is mainly due to the small pitting corrosion of the cladding layer during immersion corrosion and the weight loss caused by countless pitting corrosion. When the cladding layer is immersed, the passivation film must be corroded first, and then micropitting can be carried out. However, the corrosion of the substrate after immersion is mainly local pit-like corrosion, and the size and depth of the corrosion pits are larger, which causes more weight loss.

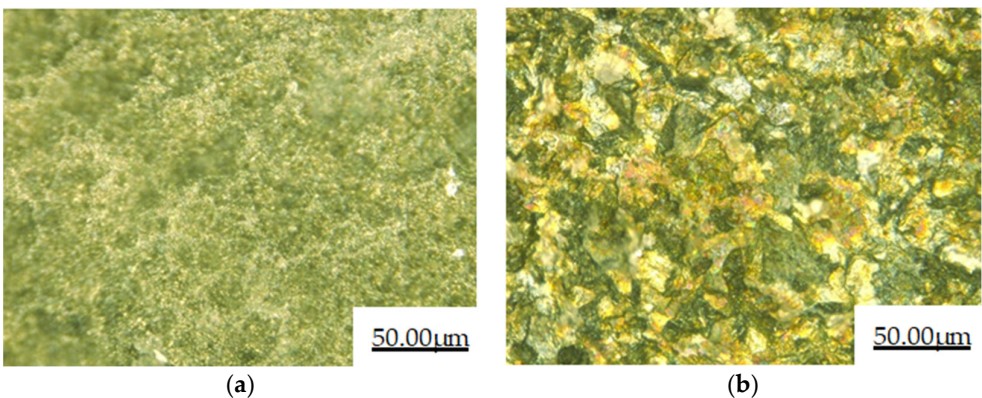

**Figure 10.** Immersion corrosion morphology of the (**a**) cladding layer and (**b**) 27SiMn substrate.

## 4. Conclusions

The Fe-based alloy coating was clad on the surface of 27SiMn by laser cladding technology to prepare a coating with a thickness of about 1.2 mm, complete morphology, a good metallurgical bond between the bottom and the substrate, and no pores or cracks in the layer. The element composition, phase composition, and structural morphology of the cladding layer were explored. Electrochemical, salt spray corrosion, and immersion corrosion studies were carried out on Fe-based alloy coatings and 27SiMn hydraulic props. The conclusions are as follow:

(1) The Fe-based alloy cladding layer is mainly composed of a-Fe solid solution, $M_7C_3$, $M_2B$, and $Cr_3Si$. Planar crystals, cellular crystals, dendrites, and equiaxed dendrites are formed in the cladding layer.

(2) The electrochemical corrosion performance test of the cladding layer and 27SiMn substrate shows that the corrosion potential of the cladding layer is larger, and the capacitive arc radius is larger than that of the substrate. This indicates that the cladding layer has better corrosion resistance than 27SiMn. The surface of the cladding layer undergoes pitting and local corrosion after electrochemical corrosion, and the surface of the substrate undergoes surface corrosion after electrochemical corrosion.

(3) The cladding layer and 27SiMn matrix were corroded by salt spray for 48 h. It can be observed that the surface of the substrate is corroded seriously, while the surface of the cladding layer sample is still smooth and has almost no change in morphology. After immersion corrosion of the cladding layer and 27SiMn substrate in the mine water simulated by the corrosive solution for 120 h, the weight loss of the cladding layer during immersion corrosion is 0.0305 $g \cdot cm^{-2}$, less than that of the substrate. The cladding layer is more resistant to corrosion than the 27SiMn substrate in salt spray corrosion and immersion corrosion.

**Author Contributions:** Investigation, data curation, and writing—original draft preparation, C.O., and Q.B.; validation, resources, project administration and funding acquisition, Q.B., and B.H.; investigation, writing—review and editing Q.B. and Y.L.; writing—review and editing, B.H. and Y.L.; formal analysis, supervision, X.Y. and Z.C. All authors have read and agreed to the published version of the manuscript.

**Funding:** This research was funded by the Natural Science Foundation of Shaanxi Provincial Natural Science Basic Research Program project (2020JM-713), Shanxi Province Key Research and Development Program (201903D121051).

**Institutional Review Board Statement:** Not applicable.

**Informed Consent Statement:** Not applicable.

**Data Availability Statement:** Not applicable.

**Conflicts of Interest:** The authors declare no conflicts of interest.

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
