# Peer review of "Microstructure and Corrosion Properties of Laser Cladding Fe-Based Alloy Coating on 27SiMn Steel Surface"

_coatings, doi:10.3390/coatings11050552_

Round 1

Reviewer 1 Report

Dear authors,

the paper “Microstructure and corrosion properties of laser cladding Fe-based alloy coating on 27SiMn steel surface” shows the influence of different corrosive environments on the cladding and base material. The topic of this paper is interesting. However, some major revisions have to be done before publishing in Coating. The main issue are the not verifiable results from XRD and the proposed growth mechanism. Furthermore, the manuscript requires a thorough spelling and grammar check.

1)    Abstract:
I)    Do not use short statements like: Objective: To improve the corrosion resistance of the 27SiMn hydraulic support. Write full sentences.
II)    The given weight losses of 0.0688 g and 0.0993 g only make sense if the surface area and the initial weight is known.

2)    Introduction: 
I)    “average particle size of 400 meshes” – please give a particle size in µm
II)    Cladding rate/(m·mn-1)should this be m·min-1?

3)    Results and discussion:
I)    The author says that a-Fe solid solution, as well as intermetallic compounds M7C3, M2B, Cr3Si and a small amount of Fe6.6Cr1.7Ni1.2Si0.2Mo0.1 compounds are found by XRD. However, this is not demonstrable given by the XRD-pattern, which is shown in the manuscript. First, all the peaks are strongly overlapping and now individual peaks are labeled for the carbide, boride, silicon precipitate phases. The only thing that is sure from XRD is that a-Fe is there. Secondly, The precipitate phases should lead to some other bragg peaks, which are not overlapping to a-fe (for example: M7C3 most probably orthorhombic). There are some unlabeled peaks at 28° and 55°. Maybe these can be related to the precipitate phases.
II)    The authors calculate the thickness of the grain by the peak broadening (Deby-Scherrer equation). This is a valid method. However, there are also other factors like instrumental broadening or internal microstrains also lead to a broadening of the diffraction peaks. It is not possible to get a reliable crystallite size in this work, as these two parameters are nor calculated. Furthermore, the crystallite size has to be rather small (usually under 1 µm) to effectively lead to a peak broadening.
III)    The authors propose that a multi-step growth process takes place during cladding that leads to different morphologies. Some of these changes are based on change in the solute concentration: “In the area above the plane crystal, as the solute concentration of the alloy gradually increases, the redistribution of the solute component will cause the component to be overcooled [14].Under the effect of the compositional supercooling zone, the unstable macroscopic flat interface transforms into a stable new interface morphology composed of many protruding round cells and network-shaped recessed grooves similar to a paraboloid of revolution. The form is called cellular crystal [15].” However, the EDS measurements show that the chemical composition in the cladding layer is almost identical to the initial powder. Therefore, the morphology change can not be explained by a change in the solute concentration.
IV)    A comparison to literature data is missing.

Author Response

Dear reviewer:

Thank you for your comment on our manuscript entitled " Microstructure and corrosion properties of laser cladding Fe-based alloy coating on 27SiMn steel surface " (ID: coatings-1198091). These comments are valuable, very helpful to revise and perfect our paper, and have important guiding significance for our research. We have carefully studied the comments and made corrections, hoping to get approval. The revised part is marked in red on the manuscript. The main corrections in this article and the responses to the reviewer’s comments can be found in the word file: Please see the attachment.

Reviewer 2 Report

The submitted manuscript deals with the corrosion resistance of iron-based coatings produced using the laser cladding technology. I have the following comments for the article:

  1. The summary is well written. It seems to be too long, but that doesn't bother me.
  2. The introduction is clear and includes relevant literature references. Maybe the author will also be interested in other papers on Fe matrix coatings (https://doi.org/10.1016/j.optlastec.2020.106784, https://doi.org/10.1016/j.jallcom.2021.159132). References only contain one item from 2021. It is useful for the paper to consider more of the most recent citations. Overall, I am of the opinion that it should be most relevant to the papers from the last 15 years. Of course, older publications are valuable as well, but they should be a small percentage of the total.
  3. Why was the sample size of 123 mm used. It is quite unusual. Was it an machine part used in mining?
  4. In Table 3 there seems to be no spaces in words "Powderfeedingrate"
  5. Figure 1 should be present in the results.
  6. Materials journal requires fill in the exact about research apparatus information in parentheses. Please check the journal requirements and match your manuscript with them
  7. Figure 2 - there are two distinct peaks, the others are less pronounced. One has been labeled (1,2,3,5), but there seems to be another peak next to it (right side). It is not very intense, but have the authors checked what it represents?
  8.  Figures 3 have not good resolution? I do not recommend correcting them, but in the future it may be worth using a different etching reagent, or taking a picture with different microscope parameters. Remove the period that follows "(c)."
  9. A great advantage of the manuscript is the use of several corrosion resistance testing techniques and their good description. There are some typos in the article, however. It is worth checking the grammatical correctness of the English. 

Author Response

Dear reviewer:

Thank you for your comment on our manuscript entitled " Microstructure and corrosion properties of laser cladding Fe-based alloy coating on 27SiMn steel surface " (ID: coatings-1198091). These comments are valuable, very helpful to revise and perfect our paper, and have important guiding significance for our research. We have carefully studied the comments and made corrections, hoping to get approval. The revised part is marked in red on the manuscript. The main corrections in this article and the responses to the reviewer’s comments can be found in the word file: Please see the attachment

Reviewer 3 Report

Dear authors,

The manuscript coatings-1198091-peer-review-v1, entitled ‘Microstructure and corrosion properties of laser cladding Fe-based alloy coating on 27SiMn steel surface’ presents the preparation of Fe-based alloy thick films onto 27SiMn steel via high-speed laser cladding method. The microstructure of the coatings, phase and element composition of the films were analyzed by the authors. The corrosion resistance of substrate and coating under different corrosion environments was analyzed by electrochemical corrosion, salt spray corrosion and immersion corrosion methods.

Characterization methods for surface investigation used by the authors are as follows: optical microscopy after metallographic corrosion, field emission scanning electron microscope (SEM) for microstructure and morphology of the sample, the chemical element distribution and content of the coating were investigated by EDS, for determine the phase of the sample an X-ray diffractometer was used.

The overall manuscript text is written well, with good explanation and descriptions.

I suggest the authors to increase the image quality and the annotation fonts size for better visualization. Also for figures 4, 5 please change the Chinese notations to English. For figures 8, 9 and 10 the scale in the figures to be made bigger for clear and easy reading, also a better contrast of the images (using a image processing software).

I propose this manuscript should be considered for publication in Coatings after meeting these suggestions.

Recommendation: MINOR Revision.

Author Response

(The authors gave the same response as above.)

Round 2

Reviewer 1 Report

The authors could answer all reviewers' questions. The quality of the paper has been improved a lot.

Reviewer 2 Report

The authors responded to all comments. Thank you for accepting them. The article should be published.